# Extracellular Vesicle (EVs) Associated Non-Coding RNAs in Lung Cancer and Therapeutics

**DOI:** 10.3390/ijms232113637

**Published:** 2022-11-07

**Authors:** Anjugam Paramanantham, Rahmat Asfiya, Siddharth Das, Grace McCully, Akhil Srivastava

**Affiliations:** 1Department of Pathology and Anatomical Sciences, University of Missouri School of Medicine, Columbia, MO 65212, USA; 2Ellis Fischel Cancer Center, University of Missouri School of Medicine, Columbia, MO 65212, USA

**Keywords:** extracellular vesicles, exosomes, non-coding (nc)RNAs, siRNA, miRNA, cancer therapeutics, lung cancer

## Abstract

Lung cancer is one of the most lethal forms of cancer, with a very high mortality rate. The precise pathophysiology of lung cancer is not well understood, and pertinent information regarding the initiation and progression of lung cancer is currently a crucial area of scientific investigation. Enhanced knowledge about the disease will lead to the development of potent therapeutic interventions. Extracellular vesicles (EVs) are membrane-bound heterogeneous populations of cellular entities that are abundantly produced by all cells in the human body, including the tumor cells. A defined class of EVs called small Extracellular Vesicles (sEVs or exosomes) carries key biomolecules such as RNA, DNA, Proteins and Lipids. Exosomes, therefore, mediate physiological activities and intracellular communication between various cells, including constituent cells of the tumor microenvironment, namely stromal cells, immunological cells, and tumor cells. In recent years, a surge in studying tumor-associated non-coding RNAs (ncRNAs) has been observed. Subsequently, studies have also reported that exosomes abundantly carry different species of ncRNAs and these exosomal ncRNAs are functionally involved in cancer initiation and progression. Here, we discuss the function of exosomal ncRNAs, such as miRNAs and long non-coding RNAs, in the pathophysiology of lung tumors. Further, the future application of exosomal-ncRNAs in clinics as biomarkers and therapeutic targets in lung cancer is also discussed due to the multifaceted influence of exosomes on cellular physiology.

## 1. Introduction

Lung cancer (LC) is the leading type of cancer and is responsible for high cancer mortality in the world [1]. Histologically LC is of two primary subtypes: Small-Cell Lung Carcinoma (SCLC) and Non-Small Cell Lung Carcinoma (NSCLC), which together make up 15% and 85%, respectively, of all LC cases. Squamous Cell Carcinoma (SCC), Adenocarcinoma (AD), and large cell carcinoma (LCC) are the other three primary forms of NSCLC. SCC and LCC, which account for 25–30% and 5–10% of all instances of lung cancer, respectively, are highly associated with cigarette smoking [2,3]. Between 40 and 45 percent of all occurrences of lung cancer are of the AD type, which is the most prevalent among both smokers and non-smokers. Such an extensive prevalence of LC has created an intense need to find effective treatment modalities for LC. However, despite an extensive effort to find a cure, the average 5-year survival rate of LC patients has been dismal at around 19% [4]. The major reason for this unfavorable response is that most lung cancer patients receive their diagnosis when the disease has already progressed to an advanced stage, making surgical resection challenging and raising the chance of recurrence after surgery [5]. Further, the response rate of chemo-, immune-, and radiation therapy with late-stage cancer is usually not successful. The unfavorable scenario is further exacerbated because, in the past, significant effort was on finding a single ubiquitous solution for all LC. After decades of studies and with the advent of novel technologies, it has now been realized that genetic heterogeneity in LC have a crucial role in disease manifestation. Therefore, the emphasis is currently on developing precision medicine approaches to treat LC, which relies on the identification of perturbation at the molecular and gene levels. Several studies have shown that in LC transcriptomic and epigenetic landscapes are significantly altered and have been explored as a resource discovering novel diagnostic and prognostic biomarkers and putative therapeutic targets [6,7]. In recent years, surveillance of bodily fluids chiefly, blood and urine have been done to detect cancer biomarkers under and collective approach of liquid biopsy [8,9]. Cell free DNA (cfDNA), Circulating Tumor Cells (CTCs) are the components of liquid biopsy that are regularly examined in liquid biopsy [10]. Extracellular vesicles (EVs)/Exosomes are the latest addition to the liquid biopsy fraternity and by several unique characteristics, have been proven superior to cfDNA and CTCs (Table 1). Exosomes have recently received much interest due to their potential significance in cancer physiology [11,12,13,14]. Both lung cancer patient’s blood and tumor tissues have been found to have unique exosome-ncRNAs signatures. As a rich source of ncRNAs exosomes actively participate in cell-to-cell communications that influence the pathophysiology of cancer cells [15,16,17]; therefore, it would be beneficial to learn how exosome ncRNA signaling is involved in different phases of cancer progression. Exosomes are also viewed as resilient therapeutic delivery vehicles and studies have been done to use them as therapeutic RNA delivery systems that have yielded robust treatment responses.

This review critically examines the de novo role of exosomal ncRNA in cancer biology and their application in cancer diagnosis and therapy.

## 2. The Identity Crisis: Extracellular Vesicles (EVs) or Exosomes

By definition, a population of sub-micron sized membrane bound vesicular structure released by cells in its extracellular environment are collectively referred to as Extracellular vesicles or EVs [13,14] Figure 1. All prokaryotic and eukaryotic cells can produce these structures in a way that has been conserved throughout evolution [15]. Exosomes, microvesicles (MVs), and apoptotic bodies are the three primary categories of EVs based on their biogenesis or biological origin [27]. MVs are membranous vesicles with a diameter of 50 to 1000 nm that may be directly budded from the plasma membrane and discharged right away [28,29,30,31]. Exosomes, which have a diameter of between 30 and 150 nm, have attracted more interest recently, nevertheless, because of their distinctive qualities. Intracellular multivesicular bodies (MVBs) and intraluminal vesicles (ILVs) are formed during several activities that result in the production of exosomes [28,29,31,32,33]. First, early-sorting endosomes (ESEs) are created because of the plasma membrane invagination [34]. ESEs eventually produce MVBs carrying ILVs via inward invagination of the endosomal limiting membrane after evolving into late-sorting endosomes (LSEs) [35,36,37]. Finally, the fusion of MVBs with the plasma membrane releases ILVs as exosomes into the extracellular environment, confirming the endosomal method of exosome formation [36,38]. The endosomal mode of exosome formation has been supported by extensive data from genetic and electron microscopy research [39]. The size, composition, biological markers, source, and function heterogeneity of EVs are all influenced by their biogenesis. Exosomes may be divided into two separate subsets: big exosomes (90–120 nm in diameter) and tiny exosomes (60–80 nm in diameter), according to combined asymmetric-flow field-flow fractionation and real-time monitoring [40]. Further, exosomes per se are very heterogenous in size and are therefore often confused by overlapping definitions of different types of secreted vesicles. To circumvent this the international Society of Extracellular Vesicles (ISEV) in 2018 published a statement paper and emphasized to classify all extracellular vesicles and the term exosomes was replaced by small Extracellular Vesicles (sEVs). However, since many of the published research discussed in this article uses the term therefore to maintain the originality of study reports we will continue to use the term ‘Exosomes ‘in this manuscript. Exosomes can be secreted by all kind of cells including macrophages, nervous cells, endothelial and epithelial cells, and mesenchymal stem cells [41,42,43,44,45,46]. One of the unique feature of the exosomes that the molecular content of exosomes chiefly consisting of various proteins, lipids, and different species of nucleic acids are representative of the cell of origin and can bring substantial changes in the recipient cells [47].Thus it is important to review the molecular content of exosomes and there influence in pathologies related to lung cancer and for that purpose this article is focus on exosomal small non coding RNA and their pathophysiological role in lung cancer.

## 3. Biogenesis and Attributes of Exosomes

The outward blebbing or budding of the plasma membrane is commonly used to generate EVs [48,49,50]. In addition to cytoplasmic pieces, apoptotic bodies frequently develop from the direct blebbing of the plasma membrane during death of a cell [51]. A range of intracellular cues, such as chemical stimulation via elevated cytosolic Ca^2+^ levels and activation of kinases that govern actin dynamics, such as the RHO GTPases, can be used to stimulate the budding followed by pinching of the plasma membrane that occurs during the production of MVs [52,53]. Exosomes, on the other hand, were shown to originate from endosomal compartments and need numerous complex mechanisms to mediate the cargo sorting and transit processes to enable apposition to the cell membrane for budding. The endosomal sorting complex needed for transport (ESCRT) machinery or ESCRT-independent pathways, such as the syndecan/ALIX pathway, ceramide, and tetraspanins, may play a role in the molecular processes driving exosome formation [54].

Based on cell types, physiological and pathological situations, as well as the manner of exosomes biogenesis, the unique contents of distinct exosomes may often indicate the unequal loading of various types of cargo [55]. We now have a growing number of publicly accessible databases with deposited information on the protein, nucleic acid, and lipid contents of exosomes, as well as the corresponding separation and purification procedures used, due to advancements in high throughput technologies like next gen DNA and RNA sequencing, mass spectrometry and high-resolution imaging [56,57,58,59]. These priceless resources will undoubtedly help in the discovery of the molecular processes behind the observed variety of exosomes cargos.

Components of the ESCRTs and proteins implicated in the numerous pathways responsible for exosomes formation, such as the RAB family of proteins, tetraspanins, as well as certain transmembrane proteins, are some of the proteins that are frequently concentrated in exosomes (Figure 1) [60]. The lipid composition of exosomes was discovered to have characteristics with that of the donor cells, with lipids including sphingomyelin, ceramide, cholesterol, and phosphatidylserine being found abundant because of their involvement in exosomes formation [61].

The nucleic acid contents of exosomes have apparently been observed to have a distinct profile from that of the donor cells, as opposed to proteins and lipids. For instance, it was discovered that some RNAs were solely concentrated in exosomes, suggesting the presence of highly selective loading mechanisms for nucleic acids [62,63,64]. Furthermore, a wide variety of nucleic acids, including pieces of mitochondrial and genomic DNA and various types of RNA have all been found in exosomes [65,66]. Valdi et al. (2007) reported exosomal shuttle RNA (esRNA) as the first nucleic acid in exosomes [65]. The successfully demonstrated the RNA from exosomes (isolated from mast cells) when added to human or mouse mast cells can show translatory effect hence they were also among the first to present the proof of concept of cell-to-cell communication via exosomes. Currently, a wide variety of RNAs like have been reported in exosomes [67,68]. However, among all types of reported RNAs non-coding RNAs present in exosomes have been studied extensively for putative role in cancer patho-biology and will be described in detail in following sections.

### 3.1. Small-Non-Coding RNA

#### 3.1.1. microRNA

An endogenous class of small non-coding RNAs is known as microRNAs (miRNAs) [69]. The bulk of miRNAs are generated through transcription of DNA sequences into primary miRNAs, precursors, and mature miRNAs [70]. miRNAs have been implicated in the processing of cancer through a variety of biological processes, including apoptosis, metabolism, and differentiation. Numerous studies have shown that miRNAs may fit into exosomes, which will help to ensure their stability and guard against degradation [71,72]. Much of the time, miRNAs cause mRNA degradation and translational repression by interacting with the 3′ untranslated region (3′ UTR) of target mRNAs [70]. To create a diagnostic screening approach for lung cancer, the exosomal miRNA profiling from plasma samples was evaluated. Exosomal miRNAs may be useful as a screening method for this tumor because the expression patterns of 12 specific upregulated miRNAs (miR-17-3p, miR-21, miR-106a, miR-146, miR-155, miR-191, miR-192, miR-203, miR-205, miR-210, miR-212, miR-214) were similar in the tumor plasma-derived exosomes and different from the control samples [73]. The exosomal miRNAs miR-378a, miR-379, miR-139-5p, and miR-200-5p have been found to be upregulated in another study as potential indicators to identify tumor from normal samples in lung adenocarcinoma. The miRNAs miR-151-5p, miR-30a-3p, miR-200b-5p, miR-629, miR-100, and miR-154-3p showed higher expression are further potential indicators to distinguish lung adenocarcinoma from granulomas [74]. More and more emphasis has recently been paid to the finding of miRNAs in exosomes. Numerous studies indicate that miRNAs are specifically sorted into exosomes, which take part in cell-to-cell interaction in the tumor microenvironment and are crucial for the biology of tumors. Exosomal miRNAs in biofluids are also readily available, abundant, and stable, making them perfect biomarkers for a variety of malignancies, including lung cancer.

#### 3.1.2. snoRNA

The introns of protein-coding or lncRNA-coding genes include nucleotide sequences called snoRNAs [75,76]. The C/D box snoRNAs and the H/ACA snoRNAs are the two primary categories of snoRNAs. Each type of snoRNA is linked to a certain gene [77,78,79,80]. The most notable number of members are found in the C/D box snoRNAs, also known as SNORD. A/C box (RUGAUGA, R = A/G), C’ box (AGUAGU), D box (AGUC), and D’ box are all contained in the SNORDs (CUGA). In eukaryotic cells, the C/D snoRNAs join with proteins like Snu13p, Nop56p, Nop58p, and fibrillarin to produce a ribonucleoprotein [81]. For certain snoRNA, no targets have yet been identified. As in the instance of piR30840, the snoRNAs are also a source of piRNAs [82]. These ncRNAs also have a special role as splicing machinery components. Some C/D box snoRNAs methylate other RNA species using their associated methylase activity, or they can establish the basis for alternative splicing, as in the instance of the snoRNA HBII-52. Like piR30840, the snoRNAs are a source of piRNAs as well [82]. These ncRNAs also have a special role as splicing machinery components. Some C/D box snoRNAs methylate other RNA species using their associated methylase activity, or they can establish the basis for alternative splicing, as in the instance of the snoRNA HBII-52 [83]. The snoRNAs have a significant impact on lung cancer. It has been established that SNORD42 has an oncogenic function in maintaining lung cancer carcinogenesis [84]. The U60, U63, U28, U51, U104, HBII-419, U59B, HBII-142, HBI-100, and U30 snoRNAs are up-regulated in lung cancer, but HBII-420 is down-regulated, according to a bioinformatic study based on TCGA data [85]. Additionally, SNORD15A expression is markedly reduced in tissue from non-smokers compared to smokers. The data also shown that non-smokers had a more uniform distribution of snoRNAs than smokers in the pattern of snoRNAs that are differentially expressed in normal tissue vs. malignant tissue [85]. In lung cancer cells, SNORD78 is overexpressed, which promotes the growth of cancerous cells, increases EMT, and thus increases their ability for invasion [86]. Furthermore, the stem phenotype of the tumor-initiating cell is maintained by these RNAs. For instance, tumor-initiating cells have low levels of the gene SNORD116-26. SNORA42 and SNORA3 are two more snoRNAs that are overexpressed in tumor-initiating cells [87]. Lung cancer cell carcinogenesis is reduced in vivo when SNORA42 is silenced in tumor-initiating cells. The prospective overall survival rate of lung cancer patients is increased by reduced expression of SNORA42 and SNORA3 in human lung cancers [87]. Exosomal Rpl13a snoRNA released from one parabiont had a 2-O-methylation impact on the rRNA in the other parabiont in an interesting recently described mouse model of parabiosis [88]. This peculiar work showed that snoRNA packing into exosomes is not just a dump of cellular byproducts but is also able to influence distal gene expression.

#### 3.1.3. piRNA

The proteins known as P-Element induced wimpy testes (PIWI) can bind with and silence transposable elements in the genome [89]. piRNAs are a subclass of short ncRNAs with 24–31 nucleotides, a bias toward the 10th adenosine or the 5′-terminal uridine, and no distinctive secondary structures [90]. To produce mature piRNA, Zucchini riboendonuclease must first cleave intermediate piRNA molecules from the piRNA cluster [91]. A piRNA/piwi complex, which plays a role in the control of spermiogenesis, germline stem cell maintenance, genomic rearrangement, transposon silencing, and epigenetic modification, is formed when piRNAs bind with the protein piwi [92]. piRNAs have so far been found in germinal cells as well as somatic tissues, including brain, heart, and plasma tissue [93]. The potential of exosome piRNAs as diagnostic biomarkers has been revealed by several research that have examined exosome piRNA profiling in a variety of disorders, including cancer, heart failure, and Alzheimer’s disease [94,95,96]. These studies help to identify healthy volunteers from patients. Cancers also exhibit dysregulation of piRNAs and piwi proteins, some of which play a role in carcinogenesis, cancer detection, and prognosis [97]. piRNAs have recently been found in exosomes of human semen, plasma, and cancer cell lines that have been cultivated [98,99,100]. It has been demonstrated that piRNAs may be transported by exosomes and influence recipient cells behavior. For instance, De Luca et al. showed that following exposure to BMSC-derived exosomes, umbilical cord blood CD34+ stem (UCB-CD34+) cells became less differentiated and more viable. Further investigation demonstrated that the miRNAs and piRNAs in exosomes control the differentiation and death of UCB-CD34+ cells [101]. In exosomes from the bone marrow of patients with multiple myeloma (MM), it was discovered that piR-004800, another piRNA, was overexpressed. This exosome-bound piR-004800 has been found to influence sphingosine-1 phosphate receptor signaling and foster the growth of MM cells, acting as an oncogene [102]. The findings with piRNAs contained within exosomes imply that these RNAs also serve the exosome-targeted cells. However, the roles played by piRNAs in exosomes have not been fully investigated. Therefore, more experimental, and clinical research is necessary.

## 4. RNA Binding Protein/Translocation of RNA to Exosomes

Most of the RNA in cells exists as ribonucleoprotein (RNP) complexes, which is an essential fact to be aware of. RBPs, or RNA-binding proteins, are crucial regulators of the post-transcriptional processing and transport of RNA molecules because of their ability to interact with RNA. RBPs control RNA processing, nucleocytoplasmic transport and maturation, intra-compartmental localization, and turnover by unique interactions with their corresponding RNA molecules [103,104,105]. RBPs exhibit the ability to interact not just with mRNAs but also with tiny ncRNAs that may create RNP complexes by binding to both single-stranded and double-stranded RNA. Such a collection of interactions is thought to be the primary regulator of various facets of cellular metabolism and can have long-lasting effects [65,106,107]. RNA-binding proteins play a key role in the sorting and loading of RNA in exosomes. The synaptotagmin-binding cytoplasmic RNA-interacting protein SYNCRIP, also known as hnRNPQ or NSAP1, has been found by Santangelo et al. as a part of the hepatocyte exosomal miRNA sorting apparatus [108]. They demonstrated that SYNCRIP knockdown reduces the ability of exosomes to internalize miRNAs. Hobor et al. later discovered that SYNCRIP has a segment known as NURR (N-terminal unit for RNA recognition), which identifies and binds the miRNA motif GGCU/A [109]. This interaction directs exosome loading of miRNA. Post-translational modification of several RBPs like YBX1 help in sorting ncRNA inro exosomes, like methylation of YBX1 aids to sort hY4F which functions as a tumor suppressor which in turn promotes lung cancer growth [110]. RNA binding proteins aid in RNA function in target cancer cells as well as RNA loading. RBP, IGF2BP1 is overexpressed in non-small cell lung cancer (NSCLC) and linked with poor prognosis (in lung adenocarcinoma) at early onset: more than 3-fold elevated IGF2BP1 expression may be identified in 70% of NSCLCs, and it positively correlates with a 30% reduction in tumor-specific 5 years survival following surgery [111]. A synergistic phenomenon between oncogenic KRAS mutation and IGF2BP1 overexpression has been documented by Rosenfeld et al. in murine and human lung cancer models [112]. KRASG12V mutant/IGF2BP1 transgenic mice showed an increased tumor development linked to KRAS mRNA attachment to IGF2BP1 [113]. Non-small-cell lung cancer cells with annexin A2 (ANXA2) knockdown had an increase in p53 and cell cycle arrest [114]. Major Vault Protein is a 100 kDa ribonucleoprotein that makes up the majority (70%) of the multimeric vault ribonucleoproteins [115]. In summary, several RNA binding proteins, alone or in conjunction with other molecular interactors, may regulate RNA sorting inside exosomes and targeting cancer mechanism.

## 5. Functional Implications of Exosome ncRNA in Lung Cancer

Exosomal ncRNAs have drawn more interest in recent years for their potential implications in the emergence of cancer. Exosomal ncRNAs have a key role in the proliferation, angiogenesis, invasion, metastasis, drug resistance, and immunological regulation of cancer cells, all crucial stages in the evolution of the disease (Figure 2). The involvement of exosomal ncRNAs in various cancer growth pathways will be further discussed in this section.

### 5.1. Cancer Progression

Cancer formation and progression are greatly aided by proliferation, which is defined by altered expression and/or activity of proteins relevant to the cell cycle. Exosomes facilitate the growth of lung cancer cells by transferring genetic information across cells in the tumor environment via exosomal miRNAs. For instance, Harel et al. discovered that exosomal miR-512 inhibited lung tumor cell growth by targeting TEA domain family member 4 (TEAD4), demonstrating that miR-512 had tumor-suppressive properties [116]. In addition, miR-208a packed in exosomes from A549 NSCLC cells was found to operate as a transfer messenger, target p21, and activate the AKT/mechanistic target of rapamycin (mTOR) pathway, limiting NSCLC cell growth [117]. Furthermore, exosomes from H1299 cells carrying miR-96 might enhance cell proliferation by specifically targeting and inhibiting LIM-domain only protein 7 (LMO7) [118].

Metastasis-associated lung adenocarcinoma transcript 1 (MALAT1) was the first lncRNA shown to be implicated in lung cancer metastasis [119]. Subsequently, Zhang and colleagues discovered that exosomal MALAT1 was substantially abundant in the serum of NSCLC patients, which sped up tumor migration and proliferation by inhibiting cell apoptosis and shortening cell cycle [120]. According to this research, exosomal MALAT1 may function as a non-invasive biomarker for the diagnosis of NSCLC or as a prospective therapeutic target for NSCLC (Table 2) [121]. Exosomal MALAT1 has a similar mechanism in many other cancers as well [122,123]. For instance, by preventing angiogenesis, GAS5 has been shown to be a possible therapeutic target for lung cancer. Human umbilical vein endothelial cells (HUVECs) are controlled in terms of apoptosis, proliferation, and tube formation by exosomal GAS5 that is generated from lung cancer. The homeobox transcript antisense intergenic RNA (HOTAIR) gene has five transcripts, all of which have been identified as lncRNAs, and is situated on the cytogenetic band 12q13.13.

### 5.2. Cancer Cell Metabolism

Exosomes have abilities comparable to those of the cells from which they originate. Exosomes carry information that exhibits traits shared by the cells that produced them. Therefore, exosomes play a role in how cancer cells metabolisms are altered. Exosomes have a key role in fostering the development of cancer by transporting enzymes, metabolites, and non-coding RNAs. For example, IGFBP4-1, a carcinogenic lncRNA discovered in lung cancer, has been shown to affect energy metabolism and encourage the production of enzymes involved in glycolysis, including HK2, LDHA, and pyruvate dehydrogenase kinase 1. (PDK1) [137]. In NSCLC, the lncRNA BCYRN1 functions as an oncogene. According to Lang et al., the miR-149/PKM2 axis was activated by BCYRN1, which also boosted PKM2 expression levels and further enhanced glycolysis in NSCLC cells [138]. CRYBG3 has recently been identified as a powerful tumor-promoting lncRNA that plays a role in NSCLC tumor spread and aneuploidy [139]. According to Chen et al., CRYBG3 and LDHA can directly interact to increase the activity of the latter, encouraging aerobic glycolysis and cell growth in NSCLC cell lines [140]. exosomal lncRNA LINC00662 promoted NSCLC progression by modulating miR-320d/E2F1 axis [141]. The reprogramming of metabolic systems by cancer cells to adapt to the stress environment might be a major factor in the observed occurrence of treatment resistance, hastening the development of cancer. Pyruvate dehydrogenase has been found to be regulated by complexes like the miR-182-PDK4 axis, which is crucial for TCA cycle and lipogenesis [142]. The downregulation of miR-22 on ATP citrate lyase (ACLY), which permitted ACLY-mediated lipogenesis and enhanced metastatic consequences, is another significant target [143]. In NSCLC, miR-126-5p can also affect MDH1’s enzymatic activity, and at higher concentrations, it can cause cell toxicity. Non-coding RNAs are crucial for understanding the beginning and development of tumors because of their function as regulators in a wide range of pathways.

### 5.3. Modifications of Immune Systems

Immune checkpoint molecules are essential for maintaining self-tolerance and preventing autoimmune disease. Patients with advanced NSCLC have had increased survival due to the targeting of immune checkpoint molecules, which are mostly represented by programmed cell death protein 1 (PD1) and its ligand PDL1. It has been shown that a network of miRNAs regulates immunological checkpoint-related functions. For instance, it has been demonstrated that miR-34, which is regulated by p53 and directly binds to the PD-L1 3’UTR to suppress its expression in NSCLC models [144]. Additionally, miR-200/ZEB1 signaling is regulated by p53. According to Chen et al., the miR-200/ZEB1 axis strongly correlates with EMT tumors and controls PD-L1 expression [145]. MiR-140 is another miRNA that binds to and restrains PD-L1. In NSCLC, miR-140 is downregulated, which increases the production of PD-L1 and, in turn, cyclin E, a gene that dysregulates the G1-S transition and the S phase in lung tumors to promote their proliferation [146]. These results collectively imply that targeting of various tumor initiating/suppressive genes by different non-coding RNAs may affect these immune checkpoints via simulating immune checkpoint inhibitor activity. Alternatively, circulating miRNAs and circRNA have been studied as potential techniques for assessing patient response to anti-PD1 treatment in NSCLCs [147]. According to Peng et al., immunological checkpoints PD1 and CTLA4 exhibit a substantial connection with the lncRNA MIR155HG, suggesting the possibility of using these models to evaluate immune inhibitors before clinical trials [148]. Denaro et al. compiled a thorough list describing the function of lncRNAs in various cancers even though no other lncRNAs have been demonstrated to have a role in immune evasion in lung cancer like HOTAIRM1 or MIR155HG [149]. Therefore, it is important to research if any lncRNAs are involved in lung cancer’s host immune system evasion.

### 5.4. Drug Resistance

One of the primary causes of the lung cancer prognosis being poor is the emergence of medication resistance. Patients with lung cancer are prone to developing resistance to molecularly targeted medications like epidermal growth factor receptor tyrosine kinase inhibitors (EGFR-TKIs), as well as to more traditional chemotherapy treatments [150]. Exploring the probable processes that reduce therapeutic effectiveness would therefore aid in developing better cancer therapies. Exosomal miRNAs had a role in EGFR-TKI resistance as well [151]. For instance, Jing et al. demonstrated the transfer of exosomal miR-21 from gefitinib-resistant H827R cells to gefitinib-sensitive HCC827 cells, where it subsequently stimulated AKT signaling and resulted in gefitinib resistance [152]. Additionally, it was noted that the levels of miR-214 were considerably greater in PC9GR cells that were resistant to gefitinib and the exosomes that were produced from those cells than in PC9 cells that were sensitive to gefitinib [153]. Gefitinib resistance developed in PC9 cells when exosomal miR-214 was transferred from PC9GR cells to PC9 cells [154]. Exosomes from PC9GR cells were transfected with an anti-miR-214, which reversed the development of gefitinib resistance.

Four exosome-derived lncRNAs have been identified so far as mediating treatment resistance in lung cancer [155]. According to one study, gefitinib exposure causes NSCLC cells to express more H19, which is transferred to other cells via exosomes produced by the main tumor [156]. Additionally, exosomal H19 targets gefitinib-sensitive tumor cells to spread gefitinib resistance. According to a related study, H19 promotes erlotinib resistance in NSCLC via the miR-615-3p/ATG7 axis. RP11-838N2.4 is another exosome-mediated lncRNA in lung cancer that promotes erlotinib resistance in NSCLC [157]. Exosomal RP11-838N2.4 is transported from erlotinib-resistant NSCLC cells to susceptible cells to induce erlotinib resistance, according to several research (Table 3).

## 6. Diagnostic Potential of Exosomes-ncRNA in Lung Cancer

Numerous studies have identified cargo ncRNAs as lung cancer biomarkers. miRNAs discovered to be dysregulated in lung cancer patients. Exosomes relative to healthy controls include miRNA-21, miRNA-23b-3p, miR-10b-5p, miRNA-139-5p, miRNA-200b-5p, miRNA-378a, miRNA-379, and miRNA-4257 [171]. Circulating miRNAs can be employed as diagnostic, prognostic, and predictive biomarkers for lung cancer, including miRNA-21, miRNA-16, and let-7. Examples of these biomarkers are miRNA-21, miRNA-122, and miRNA-205 [172]. Exosome-derived lncRNA MALAT1 may be a viable biomarker for NSCLC screening, according to a recent meta-analysis, however more validation is needed because to its limited specificity [173]. MALAT-1 levels in serum-derived exosomes are positively correlated with the tumor stage and lymphatic metastasis, according to Zang et al. [123]. Growth arrest-specific transcript 5 (GAS5), a lncRNA, was downregulated in NSCLC patients [174]. The expression of this lncRNA was inversely related to tumor stage. Exosome lncRNA HAGLR and CTCs were demonstrated to be potential biomarkers in NSCLC patients by Rao et al. in 2019 [175]. Drug resistance has also been linked to exosome lncRNAs. For instance, lncRNA H19 was connected to gefitinib resistance whereas lncRNA RP11-838N2.4 was linked to erlotinib resistance in NSCLC [176].

Serum exosomal piR-hsa-26925 and piR-hsa-5444 might be used as possible biomarkers for the diagnosis of Lung adenocarcinoma and piRNAs in lung cancer.

Liquid biopsy is a less invasive technique for examining solid tissues, blood, and other bodily fluids. Exosomes are an excellent signal for liquid biopsies since they are easily found in practically all human bodily fluids. Exosomes from various cell types and states have been found to contain unique RNA profiles, especially ncRNAs. These novel analytes offer a different method for diagnosing, tracking, and determining how effectively a treatment is working for tumor processes as well as other human illness processes, such those in viral and parasite diseases. Important cellular signaling regulators known as ncRNAs may be found and packed inside exosomes to be released into the bloodstream with great stability. Therefore, the use of exosomal ncRNAs offers intriguing potential for liquid biopsy in the case of disorders and may remain a focus of fascinating study in the area.

A few exosome databases are already being built to gather different publicly available exosome sequencing data (Table 4). High-throughput sequencing and exosome databases work well together to understand the profile of exosomal ncRNAs under certain pathophysiological settings, allowing for quick identification of exosomal ncRNAs that needs further investigation.

## 7. Exosome-ncRNA as a Therapeutic Tool in Lung Cancer

Non-coding RNAs have been used in few documented clinical studies for therapy, while being widely used as early detection biomarkers and diagnostic indicators. The first miRNA target treatment to be approved for use was MRX34, a miR-34a mimic used to treat hepatocellular carcinoma, lung cancer, and other cancers that had metastasized to the liver (NCT01829971). The viability of non-coding RNAs as possible treatments was established by this clinical experiment. A study from The Asbestos Diseases Research Institute (NCT02369198) examined the relationship between malignant pleural mesothelioma, a rare form of lung cancer, and miRNA expression. To determine if medication that modifies the regulation of non-coding RNAs would be advantageous for long-term treatment, researchers used this study to examine miRNAs from the miR-15 family and drug sensitivity. They used a synthetic miRNA called TargomiRs, especially a miR-16 mimic, to do this. Only one of the 26 participants showed any sort of partial response to the medication. Patients with NSCLC were examined with the combined chemotherapy treatment Cisplatin and Vinorelbine in different clinical research by Berghmans et al. They looked at whether mRNA and miRNA may function as prognostic biomarkers. There were no definite findings when comparing transcriptome assessments of possible miRNA biomarkers from earlier studies with actual expression in the patients. It is challenging for researchers to discover distinct biomarkers due to restrictions brought on by the diverse histology of lung malignancies and separation of miRNA expression. There will be anticipation for more clinical studies as non-coding RNA treatment in 3-D models advances. It is uncertain currently if non-coding RNAs have a role in potential treatments.

### 7.1. Exosome Mediated Delivery of Therapeutic Small RNAs

The unique properties of exosomes as a natural carrier for biomolecules, high biocompatibility and minimal systemic toxicity have now allowed them to be explored as an attractive candidate for in vivo anti-cancer therapeutic delivery system. Thus, it has been widely used as nano vectors to deliver therapeutics in medicine especially for cancer therapy [179]. Additionally, the possibility of surface functionalization of exosomes membranes with different tumor targeting moieties allows for more precise delivery targeting (Figure 3). Further, Exosomes ability to shield their payload from destruction in the extracellular environment and transport it to destination cells makes them promising candidates for precise delivery (both in vitro and in vivo) of therapeutically important RNA (most often short RNAs (siRNAs and miRNAs) targeted at treating a variety of disorders including cancer [180]. Exosomes can integrate desired therapeutic RNA directly through exogenous ways (Physical methods e.g., electroporation, sonication, freeze thaw cycle and chemical methods e.g., RNA cholesterol conjugation, Exosome and liposomal hybrid) or indirectly through endogenous methods by genetically altering the donor cells (RNA transfection, RNA encoding plasmid transfection, virus transfection, RNA and RNA binding sequence engineering) to manufacture desired RNA loaded Exosomes [181].

### 7.2. Exosomes for siRNA Delivery

The transfer of synthetic interfering RNAs via exosomes is an efficient technique for tumor RNA interference (RNAi) therapy. Bioengineered exosomes loaded with siRNAs targeting genes involved in oncogenes like BCL-2, PLK1, KRAS, the survivin protein prevents cell migration and proliferation and forms a strong foundation of exosome mediated siRNA treatment to the tumor cells. According to a study, exosomes produced by mesenchymal cells that resemble fibroblasts were designed to carry siRNA targeted at the oncogenic KrasG12D. The modified Exosomes (iExosomes) then demonstrated improved targeting to cancer-causing Kras as compared to liposomes in various pancreatic cancer mice models [182]. Wang et al. reported tumor-suppressing effects after treating hypopharyngeal carcinoma cell line (FaDu cells) with Exosomes/transient receptor potential polycystic 2 (TRPP2) siRNA conjugates [183]. In another work, an iRGD peptide produced on the surface of exosomes increased exosome-Breast cancer cell fusion and siRNA administration into tumor cells [184].

In attempt to have an enhanced therapeutic effect in lung cancer therapy exosomes have been electroporated with KRAS siRNAs and were administered to A549 tumors in vivo, KRAS was knocked down, and the tumor was subsequently suppressed. Similarly, SOX2 siRNA-engineered tLyp-1 exosomes given to NSCLC lowered proliferation and growth of cancer cells [185]. Notably, the potential of exosomes as potent carrier for siRNA is slowly being realized and a phase 1 clinical trial using MSC-derived exosomes with siRNA targeting mutant KRAS is now being conducted [NCT03608631] to treat metastatic pancreatic cancer. In a study, Lin et al. modified the surface of exosomes with PD-L1 antibody and two additional chemical moieties PEG and PEI. The Exo-PEG-PEI-PD modified targeted exosomes could target and identify tumors. The surface PEI of the modified exosomes interacted ionically to load siRNA that was specifically targeted to PD-L1. Following that, in vitro cytotoxicity and tumor cell identification and inhibition tests were performed. The findings show that the PD-L1 targeting exosome can be employed as a safe and effective nanocarrier for siRNA distribution, which is a key component of tumor-targeted gene therapy [186].

### 7.3. Exosomes as Carriers for miRNA Delivery

Like siRNA synthetically generated miRNAs can also be packaged in exosomes and delivered as an effective cancer treatment molecule. In castration-resistant prostate cancer patients, for example, miR-1290 and miR-375 in exosomes were favorably related with overall survival [187,188]. In osteosarcoma, hepatocellular carcinoma (HCC), non-small cell lung cancer (NSCLC), breast cancer, and glioma, the transport of miRNAs by MSC-derived exosomes has demonstrated promising anti-tumor effects [189]. In a recent work, modified exosomes (miR-449a Exo) that can actively distribute miR-449a were created by genetic engineering. It was demonstrated that miR-449a Exo was preferentially taken up by A549 cells and had strong homologous targeting capability. Furthermore, miR-449a Exo exhibited a good in vitro and in vivo miR-449a delivery efficiency. They showed that miR-449a Exo efficiently prevented the A549 cells from proliferating and encouraged their apoptosis. Furthermore, miR-449a Exo was discovered to slow the growth of mice tumor and lengthen the lifespan in vivo [190]. Another study examined the anti-tumor and anti-angiogenic effects of miRNAs on NSCLCs cultivated in 2D and 3D microfluidic systems.

It has also been documented that exosomes can transport miRNA inhibitors (anti-miR-9, anti-miR-214, and anti-miR-374) to cancer cells [191]. Li et al. have combined the exosomes marker protein CD9 with HuR, an RNA binding protein that has a high affinity for miR-155, to increase the loading efficiency of particular miRNAs into exosomes. The fused CD9-HuR efficiently enriches miR-155 into exosomes after cell transfection and the generation of Exosomes [192].

To transport more miRNA to tumor cells and strengthen the therapeutic effect, genetically modified Exosomes can target tumor cells by attaching functional ligands modified on exosomes surface to overexpressed receptors on the tumor surface. For instance, Liang et al. discovered that endocytosis by the SR-B1 receptor was required for the Apo-A1-modified Exosomes (Apo-Exo/miR-26a) to specifically bind to HepG2 cells. The findings showed that Apo-Exo/miR-26a-treated cells could upregulate miR-26a expression by about threefold, downregulate key cyclins CCNE2 and CDK6 expression by about onefold, and significantly inhibit cell migration by about twofold compared to HepG2 cells treated with EVs loaded with miR-26a [193]. Ohno et al. administered GE11-targeted exosomes carrying miRNA let-7a to mouse breast cancers overexpressing EGF. Exosomes with GE11 targeting displayed more tumor suppression than the control. Exosomes made from donor cells that were miRNA-transfected also reduced cancer cells [194]. Recent research has shown that chemotherapeutics and miRNAs can be co-encapsulated within modified Exosomes to produce an even better anti-tumor effect [195,196].

Additionally, Exosomes loaded with exogenous miRNAs have been tried in numerous experimental cancer treatments. EVs can also be loaded with miRNA using electroporation. According to studies, each Exosome included roughly 3000 miRNA molecules. Exo-FectTM Exosome transfection reagent, HiPerFect transfection reagent, and Lipofectamine 2000 and 3000 are a few other commercially available transfection reagents that are used to load miRNA directly into Exosomes [197,198,199,200,201,202]. Additionally, during heat shock, CaCl_2_ can facilitate the transfection of miRNAs or their inhibitors into exosomes, and these RNAs have functional activity once they have been delivered to the target cells [203]. Hyaluronic acid-polyethyleneimine (HA-PEI)/hyaluronic acid-polyethylene glycol (HA-PEG) combined nanoparticles were used by Trivedi et al. to successfully introduce miRNA125b into SK-LU-1 lung cancer cells [204]. This successfully increased miRNA-125b expression in exosome secreted by the lung cancer cells. Table 5 discussed about the recent in vivo delivery of therapeutic ncRNA by EVs.

## 8. Conclusions and Future Perspective

Exosomal non-coding RNAs are emerging as excellent liquid biopsy analytes because they can be detected in bodily fluids. However, there are several issues that must be resolved before clinical translation. The process of producing pure and homogeneous exosomes is a technological challenge worth highlighting. The methods for isolating exosomes that are now most often employed are conventional ultracentrifugation, exosome precipitation reagents, density gradient separation, immunomagnetic beads, and ultrafiltration. However, these approaches have several limitations, including a reliance on particular equipment, labor-intensive and lengthy processes, low production, and purity. As a result, there is a critical need for more accurate, trustworthy, and accessible methods for the separation and characterization of exosomes. On the one hand, these techniques may make the exosome isolation processes simpler and increase exosome production or purity. However, there is still a dearth of preclinical procedures and experience with laboratory testing. Hopefully soon, a standardized procedure for isolating exosomes will be created. Additionally, cancer is a very heterogeneous illness with various gene expression patterns in various locations and biofluids. Since various human blood fractions had varied miRNA profiles, Chen et al. revealed that certain biomarkers might only be found in a limited number of biofluids [209]. As a result, appropriate biofluids must be chosen in accordance with the features of various cancers, and they must be simple to access using minimally or non-invasive methods. 

Non-coding RNAs have mostly been the subject of clinical research in the past that were primarily concerned with using them as diagnostic markers; however, more recent clinical trials are focusing on how non-coding RNAs may be utilized as prognostic and clinical response indicators. Models that forecast dysregulation in various microenvironments may be built using artificial intelligence based on computer science. Limitations include developing a standard set of biomarkers owing to variations in cancer histology, however applying artificial intelligence might possibly source numerous datasets and identify the most efficient non-coding RNAs. Modification of microRNAs has been shown to have multiple roles. Further studies needed to ensure the ancillary effects of these ncRNA. The significance of non-coding RNAs and therapeutic treatment may be assessed and, hopefully, moved toward additional clinical trials using more sophisticated clinical models that simulate the lung tumor microenvironment.

## Figures and Tables

**Figure 1 ijms-23-13637-f001:**
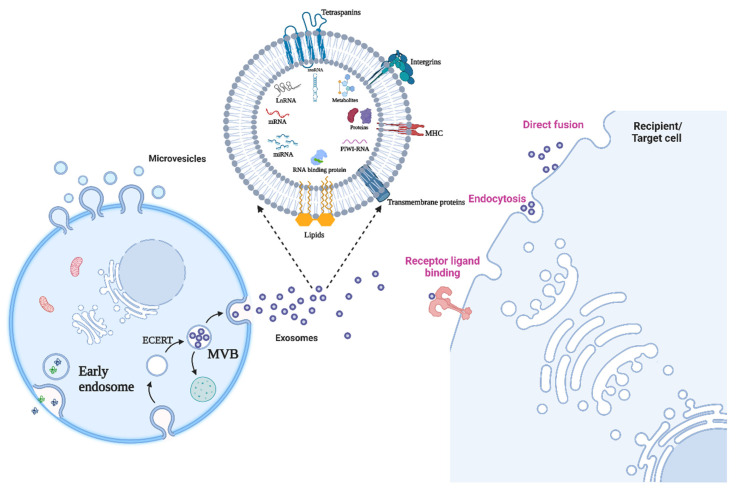
Release of extracellular vesicles and the functional impact on target cells. Since the cargo of EVs is solely reliant on the pathophysiological circumstances of the cell at the precise moment the vesicle is produced, EVs are a heterogeneous population in terms of both form and content. It is important to consider how the same vesicle, or the same message, might be interpreted differently depending on the cytotype that receives it when analyzing the intricacy of EV-mediated cell–cell communication. The recipient cell’s gene expression profile will play a significant role in this. Figure is created with BioRender.

**Figure 2 ijms-23-13637-f002:**
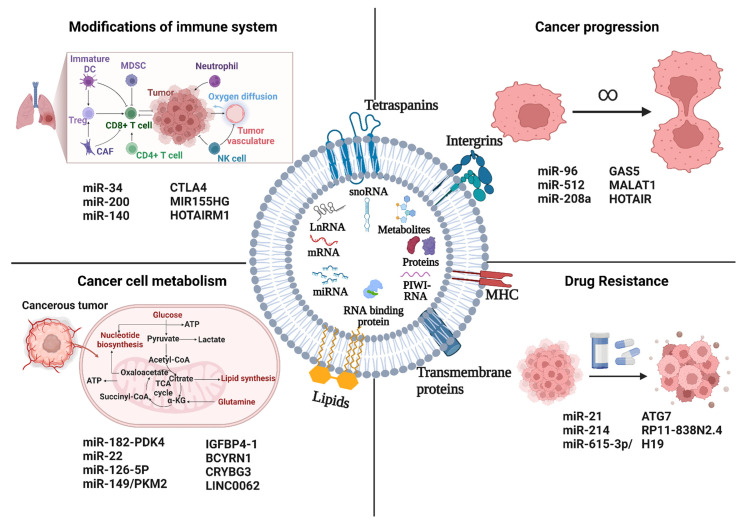
Exosomal ncRNAs and lung cancer. Malignant cells produce exosomes that transport ncRNAs that can promote tumor growth to distant organs and the cancer microenvironment. ncRNAs can have a variety of impacts on recipient cells, including: (1) Increase cancer proliferation (2) Increase cancer drug resistance; (3) Modify immune cell signaling, affecting, and changing the immune response; (4) Alter the metabolism of cancer cells. Figure is created with BioRender.

**Figure 3 ijms-23-13637-f003:**
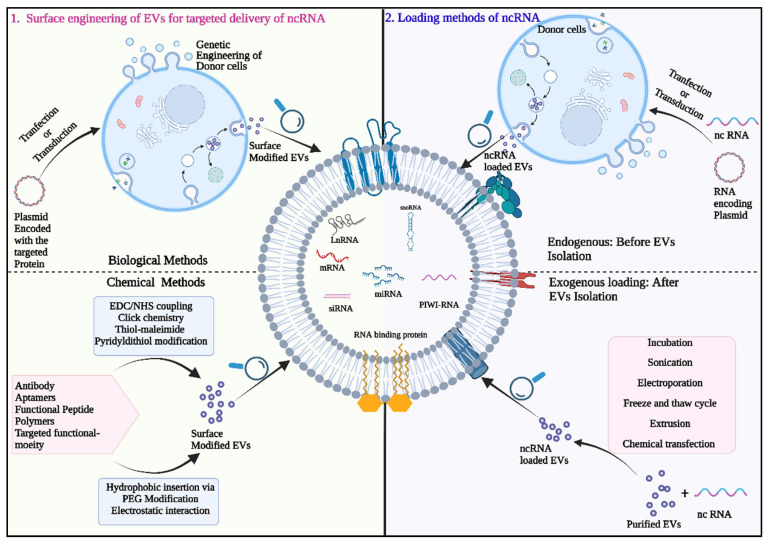
Extracellular Vesicle (EVs) for non-coding RNA (ncRNA) delivery: Overview of different ncRNA loading procedures (endogenous/exogenous loading) and methods of EVs surface modification (chemical and biological modification). Figure is created with BioRender.

**Table 1 ijms-23-13637-t001:** Comparison of the analysis capability of CTC’s, cfDNA and exosomes.

	CTC	cfDNA	Exosomes
Origin	Intact cells [18]	Necrotic/apoptotic cells/intact cells [19]	All cells [20]
Source	Peripheral blood [21]	Serum or plasma [22]	Plasma and almost all bodily fluids [20]
Early-stage detection	N/A	N/A	Detects early-stage cancer
Biomarkers	Non-coding RNA, DNA, and protein can be evaluated	Only DNA	Non-coding RNA, DNA, and protein can be evaluated
Inflammatory protein markers	N/A	N/A	Enriched with inflammatory markers, chemokines, and cytokines [23,24,25]
Clinical trials	CELLSEARCH [26]	GRAIL	UNEX-42 (NCT03857841)

**Table 2 ijms-23-13637-t002:** NcRNA’s involved in proliferation and metastasis of lung cancer.

ncRNA	Exosome Origin	Targeting Pathway	Reference
miR-23a	A549	TGF-β	[124]
miR-23a	A549	PHD1/PHD2 and ZO1	[125]
HOTAIR	A549, NCI-H1975	miR-203	[126]
UFC1	A549, H1299	PTEN	[127]
MALAT-1	A549, H1299	miRNA-491-5p/UBE2C	[128]
MMP2–2	A549	TGF-β/MMP2–2/MMP2	[129]
TBILA AGAP2-AS1	NSCLC patients	-	[130]
FOXD3-AS1	A549	ELVAL1/PI3K/Akt	[131]
LINC00662	NSCLC patients	miRNA-320d/E2F1	[132]
H19	A549	H19/miRNA-615-3p/ATG7	[133]
PCAT-1	A549	miRNA-182/217-p27/CDK6	[134]
SCIRT	A549	miRNA-665/HEYL	[135]
SOX2OT	A549	miRNA-194-5p/RAC1	[136]

**Table 3 ijms-23-13637-t003:** ncRNA’s involved in drug resistance of lung cancer.

ncRNA	Exosome Origin	Drug	Reference
miR-100-5p	A549	Cisplatin	[158]
H19	HCC827, HCC4006	Gefitinib	[159]
H19	Serum, HCC827, A549	Erlotinib	[160]
RP11-838N2.4	HCC827, HCC4006	Erlotinib	[161]
miR-425-3p	A549	Cisplatin	[162]
MSTRG.292666.16	Plasma, H1975	Osimertinib	[163]
UCA1	HCC827, PC9	Geftinib	[164]
FOXD3-AS1	A549	5-fluorouracil	[165]
AGAP2-AS1	A549, H460 and H1299	Radioresistance	[166]
cicHIPK3 cicPTK2	A549	Pexidartinib	[167]
cic0014235	Non-small cell lung cancer	Cisplatin	[168]
miR-96	H1299	Cisplatin	[169]
miR-222-3p	A549-GR	Gemcitabine	[170]

**Table 4 ijms-23-13637-t004:** Open-source curation of exosome ncRNA.

Database	Weblink	Type of ncRNA	No. of ncRNA Expressed	Reference
EVatlas	http://bioinfo.life.hust.edu.cn/EVAtlas/#/rna (accessed on 6 October 2022)	miRNA	2527	[177]
		snoRNA	1953	
		piRNA	22,546	
		snRNA	1771	
		rRNA	1294	
		tRNA	432	
		yRNA	4	
exoRBase	http://www.exorbase.org/exoRBaseV2/toIndex (accessed on 6 October 2022)	lnRNA	15,637	[62]
		mRNA	19,643	
		cirRNA	79,085	
Vesiclepedia	http://microvesicles.org/index.html# (accessed on 6 October 2022)	miRNA	10,520	[178]
		mRNA	27,646	
ExoCarta	http://exocarta.org/index.html (accessed on 6 October 2022)	miRNA	2839	[56]
		mRNA	46,879	

**Table 5 ijms-23-13637-t005:** Lists studies exploring exosomes for in vivo delivery of therapeutic ncRNA as treatment modality.

Exosomes	Source of Exosomes	Delivery Cargo	Loading Method	Target Gene	Mechanism of Loaded Therapeutics	Cancer Types (Cell Lines)	Therapeutic Effects	Reference
Engineered exosomes (iRGD peptide modified)	HEK293T cells	KRAS siRNA	Lipofectamine 2000 transfection reagent	KRAS	Knock-down KRAS gene expression	Lung cancer (A549)	tumor growth Inhibition	[205]
Engineered exosomes (tLyp-1-modifed EVs)	HEK293T cells	siR1, siR2, siR3	Electroporation	SOX2	Knock-down the SOX2 gene expression	Non-small cell lung cancer (A549)	Silenced the target gene expression and reduced the stemness of cancer stem cells	[206]
Engineered exosomes (EGFR RNA aptamer- modified)	HEK293T	Survivin siRNA	ExoFect exosome transfection	survivin	Silencing the expression of survivin	Non-small-cell lung cancer (A549)	Leading to sufficient gene silencing, chemotherapy sensitization, and regression of tumor	[207]
Engineered exosomes with PDL-1 antibody (Exo-PEG-PEI-PD)	A549 Cells	PD-L1 siRNA	Incubation	PD-L1	Silencing PD-L1 gene expression	Lung cancer (A549)	In vitro Inhibition of tumor cell proliferation and promoted the apoptosis	[186]
Engineered exosomes With TAT peptide modification	A549 cells	miR-449a	Interaction with TAT protein	Bcl-2	BCL-2 expression	Non-small cell lung cancer (A549)	Promoting cell apoptosis by inhibition of cell proliferation	[189]
Human cell-derived exosomess	HEK293T cells	Mimic of miRNA-497	Transfection	YAP1, HDGF, CCNE1, VEGF-A	Knockdown of YAP1, HDGF, CCNE1, VEGF-A expression	Non-small cell lung cancer (A549)	Angiogenesis and inhibition of tumor growth	[191]
exosomes isolated from SK-LU-1 cells	SK-LU-1 cells	miRNA-125b	Chemical transfection by Hyaluronic acid-polyethyleneimine (HA-PEI)/hyaluronic acid-polyethylene glycol (HA-PEG)	p53	Modulation of wt-p53 and miR-125b expression and reprogramed global miRNA profile for activation of pathways associated with apoptosis as well as p53 signaling	SK-LU-1 lung cancer cells	miR-449a Exo was found to control the progression of mouse tumors and prolong their survival in vivo	[203]
Lung cancer derived exosome	A549 cells	miR-563	electrophoresed	Bcl-2	Inhibiting the function of Bcl-2	Non-small cell lung cancer (A549)	miR449a significantly inhibited the expression of apoptosis inhibitor protein Bcl-2 in A549 cells and thereby promoted cell apoptosis. Tumor regression and improved survival of in vivo	[208]

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
