# Peer review of "Extracellular Vesicle (EVs) Associated Non-Coding RNAs in Lung Cancer and Therapeutics"

_ijms, 2022, doi:10.3390/ijms232113637_

Round 1

Reviewer 1 Report

From my point of view, this study was well written and gave a comprehensive overview of the diagnostic and therapeutic applications of extracellular vesicles in lung cancer. Also the sequence and transition from one topic to another was great. Also a good thing about the study was the authors' emphasis on taking into account the recommendations of the International Society of Extracellular Vesicles. I may have some comments that do not affect the quality of the manuscript. First, I think that the author used a large part of the review to talk about general and basic information not directly related to the topic of the research, but to the biology of exosomes. Another point, in the discussion section, I suggest that the authors answer two important points. First, we are at a stage where we cannot be certain about the mentioned effects of microRNA. Because most studies focus on the effect of these components on a specific tissue without ensuring the safety of the rest of the body's other tissues. Second, we cannot be certain that the components found in body fluids are exclusively present in the exosome, which makes it difficult for us to determine the reality of the effect. 

On line 544, you need to give a reference.

Authors should mention the software used to produce the images

Reviewer 2 Report

The recommendations for improvement are listed below:

1. For the RNA binding protein/translocation of RNA to exosomes, the author just mentioned one of the RNPs, SYNCRIP, there are several RNPs have been found for RNA sorting into exosomes, such as YBX1 and Lupus La. It would make this part of the summary more complete if other RNPs were added.

2. For the 3.1.1, it would be better to make a table to show which miRNA was upregulated or downregulated in exosomes from different lung cancer samples.

3. Line323-324, this sentence is not complete.
